# Adhesion to the Brain Endothelium Selects Breast Cancer Cells with Brain Metastasis Potential

**DOI:** 10.3390/ijms24087087

**Published:** 2023-04-11

**Authors:** Bai Zhang, Xueyi Li, Kai Tang, Ying Xin, Guanshuo Hu, Yufan Zheng, Keming Li, Cunyu Zhang, Youhua Tan

**Affiliations:** 1The Hong Kong Polytechnic University Shenzhen Research Institute, Shenzhen 518000, China; 2Department of Biomedical Engineering, The Hong Kong Polytechnic University, Hong Kong 999077, China

**Keywords:** endothelial adhesion, biomechanics, mechanobiology, fluid shear stress, brain metastasis

## Abstract

Tumor cells metastasize from a primary lesion to distant organs mainly through hematogenous dissemination, in which tumor cell re-adhesion to the endothelium is essential before extravasating into the target site. We thus hypothesize that tumor cells with the ability to adhere to the endothelium of a specific organ exhibit enhanced metastatic tropism to this target organ. This study tested this hypothesis and developed an in vitro model to mimic the adhesion between tumor cells and brain endothelium under fluid shear stress, which selected a subpopulation of tumor cells with enhanced adhesion strength. The selected cells up-regulated the genes related to brain metastasis and exhibited an enhanced ability to transmigrate through the blood–brain barrier. In the soft microenvironments that mimicked brain tissue, these cells had elevated adhesion and survival ability. Further, tumor cells selected by brain endothelium adhesion expressed higher levels of *MUC1*, *VCAM1*, and *VLA-4*, which were relevant to breast cancer brain metastasis. In summary, this study provides the first piece of evidence to support that the adhesion of circulating tumor cells to the brain endothelium selects the cells with enhanced brain metastasis potential.

## 1. Introduction

Tumor metastasis accounts for over 90% of cancer-related deaths and refers to the dissemination of tumor cells from a primary lesion to distant organs. This is a sequential process, in which a subpopulation of tumor cells dislodge from the primary tumor, migrate and invade the local stroma, intravasate into and survive in circulation, re-adhere to the endothelium, extravasate into distant organs, and form metastatic colonization [1]. Tumor cells must survive under a variety of rate-limiting factors and less than 0.1% of circulating tumor cells (CTCs) may eventually generate metastatic tumors, implicating the poor metastatic efficiency [2,3]. This also suggests that not all but only a small subpopulation of disseminated tumor cells have the ability to succeed in the entire metastatic process and each metastatic step may enrich a portion of disseminated cells [4]. Therefore, it is important to isolate those rare metastasis-competent tumor cells for the comprehensive characterization of their unique properties and the development of effective targeting strategies.

Tumor cells metastasize to distant organs mainly through the hematogenous route, during which CTCs need to adhere to the endothelium within the target organs before they can extravasate the vasculature. Such arrest often starts from weak adhesion between CTCs and the endothelium under blood flow and is then stabilized through a strong adhesion that is formed during the rolling of cancer cells on the endothelium [5,6]. The weak adhesion that captures cancer cells in circulation needs to be formed quickly to overwhelm the dislodgement caused by blood shear stress. It is known that selectins expressed on endothelial cells mediate such weak adhesions. The adhesion molecules on the surface of CTCs, such as CD44 and MUC1, can bind to selectins so that the weak adhesion between tumor cells and the endothelium is established [7,8,9]. To further stabilize this interaction, strong adhesion is required for the stable attachment of tumor cells to the endothelium. Integrins universally expressed on cancer cells (such as integrin α5β1 and α4β1), and their ligands on endothelial cells (such as VCAM-1 and ICAM-1), have been reported to mediate this strong adhesion [6,10,11,12]. Some of these adhesion molecules involved in either weak or strong adhesion between tumor cells and endothelium, such as CD44, VCAM1, and VLA4, play indispensable roles in metastasis [12,13,14,15]. This adhesion process may inevitably select CTCs with certain adhesion molecule profiles and thus enrich a subpopulation of these cells with enhanced metastatic potential.

Tumor cells do not randomly choose their metastatic sites; rather, they disseminate to the preferred organs for metastatic colonization, which is referred to as organotropism [16]. In particular, endothelial cells in different organs may exhibit distinct expression profiles of adhesion molecules, which may be critical in the re-attachment of tumor cells to the specific endothelium and in mediating metastatic organotropism in the target site [17,18]. However, it remains unclear whether the tumor cell-endothelium adhesion can select a subpopulation of cells that preferentially metastasize to a specific organ. In this study, we developed a microfluidic system to mimic the interaction between CTCs and endothelial cells in the brain tissue. Tumor cells that stably adhered to the brain endothelium under fluid shear stress were enriched and their brain metastasis ability was characterized, including the gene expression profile, adhesion to the brain endothelium, transmigration ability across the blood–brain barrier (BBB), and adhesion and survival in the brain tissue. In addition, the expression profile of adhesion molecules in these selected cells was analyzed and their clinical relevance was examined.

## 2. Results

### 2.1. In Vitro Selection of Breast Cancer Cells through the Adhesion to the Brain Endothelium

Tumor cells must adhere to the brain endothelium before extravasating into the tissue to generate brain metastases. To mimic the interaction between the endothelium and CTCs, we developed an in vitro system (Figure 1a) [19,20,21], in which a monolayer of brain endothelium (hCMEC/D3) was grown in the microfluidic channel and tumor cells in suspension were circulated under a steady flow in the system. Since CTCs are very rare in vivo (1–10 CTCs per mL blood) and not commercially available yet, tumor cells in suspension have been widely utilized as an alternative model for CTC study [22,23,24,25,26]. In this study, suspended MDA-MB-231 cells (WT) were used to mimic breast CTCs, and they expressed high levels of the CTC marker EpCAM with high metastatic potential [27,28]. The endothelial monolayer was formed without the pre-treatment under shear stress before the perfusion of suspended tumor cells. The adhesion of tumor cells to the endothelium could be allowed in the blood flow within the veins (0.5–4 dyn/cm^2^) while being prevented in the arteries (4–30 dyn/cm^2^) [5]. The shear rate of blood flow in the brain post-capillary venules varies within a range of 1–6 dyn/cm^2^ [29,30,31]. To determine the suitable fluid shear stress level, wide type MDA-MB-231 cells (WT) were perfused at different shear stress (5, 2.5, 1, 0.5, and 0.25 dyn/cm^2^) into the flow chamber for 15 min and the adhesion between tumor cells and endothelial cells was analyzed. The results show that a large number of WT cells adhered to the brain endothelium under 0.25 dyn/cm^2^ wall shear stress (Figure 1b,c). However, the adhered cells considerably decreased when the shear stress increased to 0.5, 1, 2.5, and 5 dyn/cm^2^. There was no significant difference in the number of adhered WT cells among the larger shearing groups (Figure 1c). We speculated that there existed a threshold between 0.5 dyn/cm^2^ and 0.25 dyn/cm^2^, which was strong enough to counteract the adhesion between cancer cells and the endothelium. Therefore, 1 dyn/cm^2^ shear stress was chosen for the in vitro brain-endothelium-adhesion-based selection. The selected cells were defined as the “Flow Adhesion Selected” (FAS) group. To explore the effect of shear stress on the endothelium-adhesion-based selection, WT cells were suspended in the flow chamber to interact with the brain endothelium for 30 min without exposure to shear stress. The isolated tumor cells were defined as the “Static Adhesion Selected” (SAS) group. Notably, almost all of FAS and SAS cells expressed the green fluorescence, indicating that they were mainly green fluorescent protein (GFP)-labeled tumor cells and free of endothelial cells (Appendix A). To compare the adhesion ability of FAS and SAS groups, both groups were mixed with WT and allowed to interact with the brain endothelium under 1 dyn/cm^2^ wall shear stress. The results show that the FAS group exhibited enhanced adhesion to the brain endothelium compared to both the SAS and WT groups (Figure 1d,e). Meanwhile, there was no difference between the SAS and WT groups. These findings suggest that the in vitro endothelium adhesion can select a subpopulation of tumor cells.

### 2.2. Adhesion-Selected Tumor Cells Up-Regulate Brain Metastasis Genes and Exhibit Enhanced Adhesion Strength and BBB Transmigration Ability

CTCs need to adhere to the endothelium before extravasation, indicating the importance of endothelial adhesion in tumor metastasis [32,33]. Therefore, we hypothesized that the endothelial adhesion might select a subpopulation of tumor cells with brain metastatic advantages. To test this hypothesis, the gene expression profile of the selected cells was examined. The results show that compared to both WT and SAS groups, the FAS group up-regulated *COX2*, *EREG*, *HBEGF*, *ITGAV*, *ITGB3*, *ANGPTL4*, *PIEZO2*, *SCNN1A*, and *LTBP1* (9 out of 11 genes; Figure 2a), which were reported to be highly expressed in breast cancer cells with brain metastasis ability [34,35,36,37]. In contrast to other metastasis sites, one special challenge for breast cancer cells in arriving at the brain is to cross the BBB, which can be mitigated by multiple key molecules, such as COX2, EREG, HBEGF, ITGAV, ITGB3, ST6GALNAC5, ANGPTL4, PIEZO2, and SCNN1A [34,35,37,38,39,40]. In particular, the FAS group showed higher expressions of *COX2* and *ITGB3*. COX2 can enhance the permeability of BBB by up-regulating the expression of matrix metallopeptidase 1 (MMP1) [37], and integrin αvβ3 (encoded by *ITGAV* and *ITGB3*) facilitates breast cancer brain metastasis by mediating the adhesion between cancer cells and brain endothelium and up-regulating the expressions of MMP2 and MMP9 [41,42]. We further examined the expressions of these two molecules at the protein level (Figure 2b–e and Appendix A). Compared to both WT and SAS groups, the FAS group enhanced the expressions of COX2 and integrin αvβ3. Note that there was no significant difference in the mRNA expressions of *ST6GALNAC5* and *SerpinB2* (Figure 2a). ST6GALNAC5 is a specific mediator of brain metastasis [34]. However, this gene might not be an indispensable biomarker for all breast cancer cells. It is found that overexpression of ST6GALNAC5 in MDA-MB-231 cells hinders their adhesion to the BBB [43], which may partially explain why MDA-MB-231 cells selected through adhesion (FAS) do not exhibit enhanced ST6GALNAC5 expression. ANGPTL4, PIEZO2, SCNN1A, LTBP1, and SerpinB2 promote the survival of breast cancer cells in the brain microenvironment [33,34,35,40]. All five genes except SerpinB2 were up-regulated in the FAS group (Figure 2a). Interestingly, FAS had enhanced protein expression of SerpinB2, indicating that the influence might be post-transcriptional. To elucidate the roles of individual molecules, it will be important to silence these genes and investigate the potential effects on the adhesion to the endothelium and the metastatic process.

Once captured by the endothelium, cancer cells begin to roll on it in the direction of the blood flow. During this process, complex molecules on tumor cells, such as integrins, can mediate strong adhesions to stabilize the interaction between tumor cells and endothelial cells [6]. To test whether FAS cells had advantages in establishing strong adhesion to the endothelium, different tumor cells were first co-cultured with the brain endothelium for 30 min to allow adhesion molecules to interact with each other. Then, different levels of shear stress were utilized to test the adhesion strength (Figure 3a,b and Appendix A) [44]. The results show that the number of adhered tumor cells decreased when the applied shear stress increased from 0.25 dyn/cm^2^ to 2 dyn/cm^2^ (Figure 3a,b). The FAS group had more tumor cells remaining on the brain endothelium compared to SAS and WT groups (Figure 3a,b). Interestingly, all three groups had a similar number of adhered tumor cells after exposure to wall shear stress higher than 2 dyn/cm^2^ (Appendix A). These results suggest that FAS cells can develop stable adhesion to the endothelium with enhanced strength.

After the stable attachment to brain endothelium, CTCs may start to transmigrate through the BBB and extravasate into the brain tissue [1,33]. To test the transendothelial migration, the brain endothelial monolayer was cultured on the top of the transwell membrane (Figure 3c) [45]. The number of tumor cells transmigrated to the bottom of the membrane was then counted. The results show that the FAS group had more than two times the number of tumor cells transmigrating through the hCMEC/D6 monolayer compared to the WT and SAS groups (Figure 3d,e), indicating the enhanced BBB transmigration ability of FAS cells. This might be related to the enhanced expressions of *COX2* and *ITGB3* (Figure 2a,b,d and Appendix A).

### 2.3. Adhesion-Selected Tumor Cells Exhibit Advantages in Cell Adhesion and Survival within the Soft Brain Microenvironment

After the transmigration across BBB, tumor cells invade into the brain tissue and need to adhere to the parenchyma and survive in the brain microenvironment. Soft polyacrylamide gels (0.6 kPa) coated with collagen I were utilized to mimic the soft environment of brain parenchyma [46]. To explore the adhesion ability of tumor cells on the soft tissue, WT cells were mixed with FAS or SAS cells equally. The cell mixture was then co-cultured on soft gels. After 15 min, the gels were rinsed to remove tumor cells with weak adhesion (Figure 4a). The results show that compared to both SAS and WT groups, the FAS group had around two-fold more cells remaining on the soft gels (Figure 4b,c), suggesting that FAS cells have advantages in adhering to the soft brain tissue.

To test their ability to adapt and survive in a soft brain environment, the morphology, survival, and proliferation of tumor cells were examined. FAS cells showed a much lower cell apoptosis than WT and SAS cells in the soft microenvironment (Figure 4d,e), while there was no significant difference between the WT and SAS groups. This suggests that FAS cells may have survival advantages in the brain tissue after extravasation. The morphology analysis shows that there was no difference in cell-spreading area, circularity, and aspect ratio among the three groups when tumor cells were cultured on tissue culture plates (Appendix A) [47,48]. On the soft gels, FAS exhibited a relatively larger spreading area than SAS cells but not larger than that of WT cells and a moderately higher aspect ratio than both the WT and SAS groups (Appendix A). In addition, FAS cells proliferated faster than WT cells but not SAS cells and showed a similar migration ability on soft matrices compared to the other two groups (Appendix A). All these results suggest that FAS cells may adapt better to soft microenvironments.

### 2.4. The Selected Tumor Cells Up-Regulate Adhesion Molecules That Are Correlated with Breast Cancer Brain Metastasis

To better understand the molecular mechanism underlying the enhanced brain metastatic abilities of the FAS cells, we examined the profile of the well-known adhesion molecules, which are reported to be highly involved in breast cancer brain metastasis [14,33,34]. The results show that compared to other groups, the FAS group had significantly higher mRNA expressions of multiple adhesion molecules, including MUC1, VCAM1, and VLA-4 (Integrin α4β1, encoded by *ITGA4* and *ITGB1*) (Figure 5a). The mRNA expressions of *MUC1* and *VCAM1* were notably up-regulated. We thus examined these two adhesion molecules at the protein level. Consistently, FAS cells expressed enhanced levels of both MUC1 and VCAM1 compared to WT and SAS group (Figure 5c–f). MUC1 interacts with selectins expressed on the surface of endothelial cells. Such adhesion forms fast and captures CTCs from the blood flow [8]. VCAM1 binds VLA-4 and both of them are expressed in tumor cells and brain endothelial cells. Their adhesion develops during the rolling of tumor cells on the endothelium and the binding strength is relatively strong [13,14]. This indicates that FAS cells might be selected by both transient capturing and the subsequent rolling process. In addition, the FAS cells up-regulated integrin β1 (encoded by *ITGB1*) that specifically interacts with collagen I, which might play a role in the enhanced adhesion and survival in soft microenvironments coated with collagen I (Figure 4). To further test the involvement of these adhesion molecules in breast cancer brain metastasis, we compared the expressions of these molecules using patient data collected from the Gene Expression Omnibus (GEO) database. The results show that *MUC1* and *ITGB1* tended to have higher expression levels in brain metastases compared with primary breast tumors (Figure 5b). Interestingly, the expression levels of *VCAM1* and *ITGA4* decreased in brain metastases compared to primary tumors. Together, these results indicate that the FAS cells up-regulate multiple adhesion molecules, which are clinically relevant in breast cancer brain metastasis.

## 3. Discussion

Tumor cells disseminate to distant organs mainly through hematogenous metastasis, in which CTCs need to re-adhere to the endothelium before extravasation and the establishment of metastatic tumors. Therefore, it is rational to hypothesize that the adhesion to the endothelium can enrich a subpopulation of tumor cells with metastatic competence. This study focused on breast cancer brain metastasis, which has a relatively high incidence, poor prognosis, and short survival time [49,50,51]. A subpopulation of breast cancer cells were isolated based on their adhesion to the brain endothelium under shear stress. These selected breast cancer cells (FAS) up-regulated multiple genes related to brain metastasis and exhibited advantages of brain metastasis, including enhanced cell adhesion to the brain endothelium, elevated transmigration through BBB, and increased adhesion to soft brain tissue and reduced apoptosis within the soft brain microenvironment. It is known that fluid shear stress influences the expressions of adhesion molecules on endothelial cells [52,53,54]. In this study, the endothelial monolayer was not pre-treated under shear stress before tumor cells were perfused, which might affect the levels of surface adhesion proteins on endothelial cells and thus the selection of FAS cells. To better recapitulate the in vivo condition, brain endothelial cells will be pre-treated under shear stress mimicking the brain blood flow at different flow rates in the future (1–6 dyn/cm^2^) [29]. The adhesion molecule profiles of the brain endothelial cells after shear treatment and the FAS cells selected by these pre-treated brain endothelial cells will be characterized.

Our findings suggest that the selected tumor cells (FAS) may be competent in generating brain metastases. Notably, breast cancer cells selected through adhesion to the brain endothelium without exposure to shear stress (SAS) did not obtain enhanced brain metastasis abilities and exhibited barely any difference from the wild-type cancer cells. This sheds light on the indispensable role of both cell adhesion and blood shear stress in metastasis. Further, several adhesion molecules were highly expressed in the FAS cells and clinically relevant to breast cancer brain metastasis. Therefore, this study provides the first piece of evidence to demonstrate that the re-attachment to the brain endothelium enriches CTCs with brain metastatic potential. This is consistent with the previous finding that cervical cancer cells selected through the adhesion to the endothelium under fluid shear stress for 48 h exhibit a high metastatic potential [20]. The influence of this selection process may involve the potential effect of long-time exposure (48 h) to fluid shear stress, while the short-time selection (15 min) in this study mainly reflects the influence of tumor cell–endothelium adhesion on brain metastasis ability. On the other hand, previous studies show that tumor cells strongly adhered to the underlying substrates are less migratory, while the cells with low adhesion strength have enhanced metastatic potential [55,56]. In addition, our results also support that the adhesion to the endothelium is an important rate-limiting factor in determining the metastasis inefficiency. Despite the adhesion molecules investigated in this study, many other proteins are known to involve in the adhesion of CTCs to the endothelium, such as VE-cadherin and N-cadherin [57,58]. In the future, proteomic analysis will be conducted to comprehensively characterize the adhesion molecule profile of FAS cells [59], which may identify the target adhesion molecules. Furthermore, the role of each target adhesion molecule will be elucidated by blunting its function using blocking antibodies and testing tumor cell adhesion and the functions of the selected cells, which can be further utilized to identify druggable targets. Thus, the results in this study suggest a possible therapeutic strategy—targeting the adhesion molecules on CTCs may prevent their re-adhesion and thus suppress tumor metastasis at the relatively early stage of tumor progression [13,15,49,60].

It is well-known that different types of cancer do not randomly metastasize to other organs; instead, they have preferred distant sites or exhibit organotropism [61]. For example, most prostate cancer metastasizes to bone and pancreatic cancer often metastasizes to the liver [62,63]. Breast cancer mainly disseminates to the bone, liver, brain, and lung [51]. The underlying mechanism remains unclear. In particular, it is largely unknown whether metastatic organotropism is related to the expression profiles of adhesion molecules in tumor cells and endothelial cells of specific organs. In this study, we report that the FAS cells highly express MUC1, VCAM1 and VLA-4. Meanwhile, the brain endothelial cells are reported to have relatively high expressions of their corresponding ligands: E-selectin, VLA-4, and VCAM1 [14,64]. These suggest that the adhesion molecule profiles of FAS cells and brain endothelial cells may match with each other. Previous studies show that endothelial cells from different organs are heterogeneous and distinct in their adhesion molecule profiles [17,18], which may arrest different subpopulations of CTCs with different organotropism. Further, the blood flow pattern of each metastasized organ varies, including the flow velocity and shear stress [5], which may affect the adhesion process between CTCs and the endothelium. Therefore, it is reasonable to assume that the endothelium-adhesion-based selection might contribute to organotropism. In the future, endothelial cells originating from bone, liver, lung, and brain will be used to select breast cancer cells under fluid shear stress mimicking the hematogenous pattern of the corresponding organ. In addition, proteomic analysis will be conducted to characterize the adhesion molecule profiles and gene signatures of the tumor cells selected through the adhesion to the endothelial cells of different organs. Finally, the metastatic preference and adhesion molecule profiles of selected breast cancer cells would be rigorously characterized.

## 4. Materials and Methods

### 4.1. Cell Culture

Human breast cancer cell line MDA-MB-231-TGL (WT) and human brain endothelial cell line hCMEC/D3 were purchased from Memorial Sloan Kettering Cancer Center and ATCC (Manassas, VA, USA), respectively. WT was a stable cell line by transfecting MDA-MB-231 cells with human herpesvirus 1 TK, EGFP, and firefly luciferase. WT cell line and its derivatives in this study were all cultured in Dulbecco’s Modified Eagle Medium (DMEM; HyClone, Logan, UT, USA) with 10% fetal bovine serum (FBS; HyClone) and 1% penicillin/streptomycin (PS; Gibco, Dublin, Ireland) at 37 °C and 5% CO_2_. hCMEC/D3 cells were cultured in Endothelial Cell Medium (ECM; ScienCell Research Laboratories, Carlsbad, CA, USA) with 10% FBS (ScienCell) and 1% PS (ScienCell), and 1% endothelial cell growth supplement (ScienCell) at 37 °C and 5% CO_2_. All cell lines were passaged every 2 to 3 days using trypsin-EDTA solution (HyClone).

### 4.2. Isolation of Flow-Adhesion-Selected Cells (FAS) and Static-Adhesion-Selected Cells (SAS)

In the microfluidic system, a brain endothelial monolayer was cultured on flow chamber slides (µ-Slide I Luer, Cat. No: 80176, ibidi, Martinsried, Germany), while the shear stress was generated using a peristaltic pump (P-230, Harvard Instruments, Cambridge, MA, USA) to mimic the blood flow. In short, the slide channels were first coated with 0.2 mg/mL rat-tail collagen type I (Thermo Fisher Scientific, Waltham, MA, USA). Then, 100,000 hCMEC/D3 cells were seeded in the slide channel for 2 days to reach high confluency. For the selection of FAS cells, 100,000 WT cells (EGFP labeled) were suspended in DMEM full medium and then infused into the flow chamber under 1 dyn/cm^2^ wall shear stress for 15 min following the protocol as previously described [20,21]. The adhered tumor cells and hCMEC/D3 cells were harvested and cocultured for 2 days in tissue culture plates (TCP). Since all tumor cells were labeled with EGFP, the selected tumor cells could be separated from the unlabeled hCMEC cells based on the fluorescence expression using BD FACSAria III Cell Sorter (BD Bioscience, San Jose, CA, USA). To test the purity of these selected cells, the expression of EGFP was then examined under a fluorescence microscope (Leica, Wetzlar, Germany). For the selection of SAS cells, an hCMEC/D3 monolayer was formed; then, 100,000 WT cells were added into the flow chamber and allowed to interact with the endothelium without exposure to shear stress for 30 min. Non-adhered tumor cells were removed via gentle washing with PBS (HyClone) and the adhered cells were isolated using the same method as FAS. The selected cells were cultured in TCP for at least three passages before experiments. They were detached using 0.2% EDTA solution (HyClone) and passaged every 3 days.

### 4.3. Quantitative RT-PCR Analysis

The total RNAs of each sample were extracted using Aurum Total RNA Mini Kit (Bio-Rad, Hercules, CA, USA). The complementary DNA was synthesized from the RNA samples using the RevertAid RT Reverse Transcription Kit (Thermo Fisher Scientific). The Forget-Me-Not qPCR Master Mix Kit (Biotium, Fremont, CA, USA) was used to prepare the PCR mixture for the quantitative analysis via the CFX96 Real-Time System (Bio-Rad). The relative expression of target genes was normalized to the expression of the housekeeper gene GAPDH. All primers used in the qPCR analysis were designed based on the National Center for Biotechnology Information database (NCBI; Bethesda, MD, USA) and listed in Appendix A.

### 4.4. Transendothelial Migration Assay

The transendothelial migration assays were carried out using Corning^®^ Transwell^®^ (Corning, NY, USA) with an 8 μm pore. In brief, the transwell inserts were coated with collagen type I (Thermo Fisher Scientific) and then seeded with 20,000 hCMEC/D3 cells. These cells were then incubated for at least two days to form a monolayer. A total of 50,000 cancer cells were then marked with cell tracker CytoTrace™ Green CMFDA (Thermo Fisher Scientific) following the manufacturer’s instructions and added into the inserts containing low-serum medium (DMEM + 1%FBS + 1%PS). DMEM full medium was then added to the lower chamber, inducing the cancer cells to transmigrate through the brain endothelial monolayer. After 24 h, the transmigrated cancer cells were fixed with a 4% paraformaldehyde solution (Thermo Fisher Scientific) and counted under a fluorescence microscopy. For each insert, microscopic images of the transmigrated cells were taken at five random views, and then the number of cells was calculated using the particle analysis in ImageJ 1.46 (NIH, Bethesda, MD, USA).

### 4.5. Flow Adhesion Assay

An hCMEC monolayer was formed in the flow chamber slides (ibidi) as described above. Then, two groups of cancer cells were marked with two different colors of cell trackers CytoTrace™ Green CMFDA (Thermo Fisher Scientific) and PKH26 Red Fluorescent Cell Linker (Sigma-Aldrich, Saint Louis, MA, USA) following the manufacturer’s instructions. In total, 100,000 cells from each group were mixed and then infused into one slide under 1 dyn/cm^2^ shear stress for 15 min. The number of adhered cells was counted.

### 4.6. Static Adhesion Assay

The hCMEC monolayer was formed in the flow chamber slides (ibidi). A total of 300,000 cancer cells were marked with cell tracker CytoTrace™ Green CMFDA (Thermo Fisher Scientific) following the manufacturer’s instructions and then added into the slides. The slides were then incubated at 37 °C for 30 min to allow cancer cells to interact with the endothelium. DMEM full medium was perfused through the slide, creating certain level of shear stress for 15 min. The number of adhered cancer cells was counted.

### 4.7. Preparation of Polyacrylamide Hydrogels (PA-Gel)

The PA-gels were synthesized and pretreated using methods reported elsewhere [46]. In brief, 40% acrylamide solution (Bio-Rad), 2% bis-acrylamide solution (Bio-Rad), and water were mixed in the ratio of 179:15:6 (0.6 kPa). Then, 1% *v*/*v* ammonium persulfate (APS, Thermo Fisher Scientific) and 0.1% *v*/*v* methylethylenediamine (TEMED, Thermo Fisher Scientific) were added into the mixture. An adequate amount of the mixed solution was added onto the chloro-silanated glass surface and then covered by amino-silanated coverslips. After solidification, the PA gels stuck to the coverslip were detached from the glass together for later use. Before assays, the PA gel was coated with 200 μg/mL collagen type I (Thermo Fisher Scientific) via the crosslinker sulfosuccinimidyl 6-(4′-azido-2′-nitrophenylamino) hexanoate (sulfo-SANPAH; Thermo Fisher Scientific).

### 4.8. Adhesion Assay on PA-Gel

A total of 100,000 cells of WT group marked with CytoTrace™ Green CMFDA (Thermo Fisher Scientific) according to the manufacturer’s instruction and 100,000 cells of the SAS group or FAS group marked with PKH26 Red Fluorescent Cell Linker (Sigma-Aldrich) were mixed and seeded on the 0.6 kPa PA gel and then incubated at 37 °C for 15 min. The cells were rinsed gently with PBS. The adhered cells remaining on the gels were counted.

### 4.9. Cell Viability Assay

A total of 300,000 cancer cells were seeded on each 0.6 kPa PA-gel using low-serum medium (DMEM + 1%FBS + 1%PS) and then incubated at 37 °C for 24 h. All cells were collected and stained using propidium iodide (PI; Abcam, MA, USA) following the manufacturer’s instructions. The ratio of dead cells to live cells was examined using a flow cytometer (BD Accuri C6, BD Bioscience). The data was analyzed using the software FlowJo_v 10.6.2 (BD Bioscience).

### 4.10. Cell Morphology Analysis

Cancer cells were either seeded on PA-gels or TCP and then incubated at 37 °C for 24 h. Microscopic images of cells were captured under bright fields. ImageJ (NIH, Bethesda) was used to mark the cell edge and measure the spreading area, aspect ratio and circularity of each cell. At least 100 cells were analyzed for each group.

### 4.11. Cell Proliferation Assay

A total of 300,000 cancer cells were seeded on 0.6 kPa PA-gels and then incubated overnight. The EdU (5-ethynyl-2′-deoxyuridine) kit (Beyotime, Shanghai, China) was used to mark and quantify the proliferating cells following the manufacturer’s instructions. In brief, cancer cells were cultured in full medium containing 0.1% EdU for 2 h. These cells were then collected, fixed, and then permeabilized. A click reaction solution from the kit was then used to label the EdU with fluorescence dyes. The percentage of proliferating cells (EdU^+^) was examined through flow cytometry.

### 4.12. Wound Healing Assay

A culture insert (Ibidi, Gräfelfing, Germany) with two separate chambers was first placed on 0.6 kPa PA-gel. A total of 200,000 cells were seeded into each chamber of the insert and incubated at 37 °C for 6 h to adhere to the PA gel. The culture insert was then removed, thus creating a wound between cells cultured in two chambers. These cells were incubated at 37 °C for another 24 h. In total, 4× microscopic images of the wound were recorded at 0 h and 24 h. ImageJ was used to mark and measure the wound area and then calculate the wound healing rate, which is the healed area (wound area at 0 h − wound area at 24 h) divided by the wound area at 0 h.

### 4.13. Immunofluorescence Staining

Cells were seeded on collagen-I-coated glass-bottom dishes (Ibidi) overnight. They were fixed with 4% formaldehyde (Sigma-Aldrich) for 15 min and then permeabilized with 0.1% Triton X-100 (Sigma-Aldrich) in PBS containing 1% BSA. Then, the cells were incubated with the primary antibodies (AbCam) of COX2 (ab188183), SerpinB2 (ab47742), VCAM1 (ab134047), MUC1(ab109185), and integrin αvβ3 (ab7166) at 4 °C overnight. After that, the cells were incubated with secondary antibodies (ab150079; ab150115) in the ark for one hour. Finally, the cells were stained with Dapi (Thermo Fisher Scientific) to counterstain the nuclei. For each group, at least 50 cells were imaged using fluorescence microscopy (Nikon, Tokyo, Japan). ImageJ was used to mark the cell boundary and measure the mean fluorescence intensity.

### 4.14. Statistical Analysis

All data in this project were shown as mean ± SEM. One-way ANOVA was adopted when there were three or more groups of samples. In the ANOVA tests, Tukey or Bonferroni tests were used to conduct the multi-comparison between every two paired groups with equal or unequal sample sizes. *, *p* < 0.05, **, *p* < 0.01, ***, *p* < 0.001, ****, *p* < 0.0001.

## 5. Conclusions

This study utilized an in vitro adhesion system to select a subpopulation of CTCs that had the ability to adhere to the brain endothelium under blood shear stress. The selected tumor cells expressed increased levels of brain metastasis genes and exhibited brain metastasis competency, including the ability to transmigrate through the blood–brain barrier and adhere/survive in the soft brain microenvironment. Further, the adhesion-selected cells up-regulated the adhesion molecules that were relevant to breast cancer brain metastasis. In summary, these findings demonstrated that the adhesion of CTCs to brain endothelium under shear stress could enrich a subpopulation of tumor cells with brain metastasis ability.

## Figures and Tables

**Figure 1 ijms-24-07087-f001:**
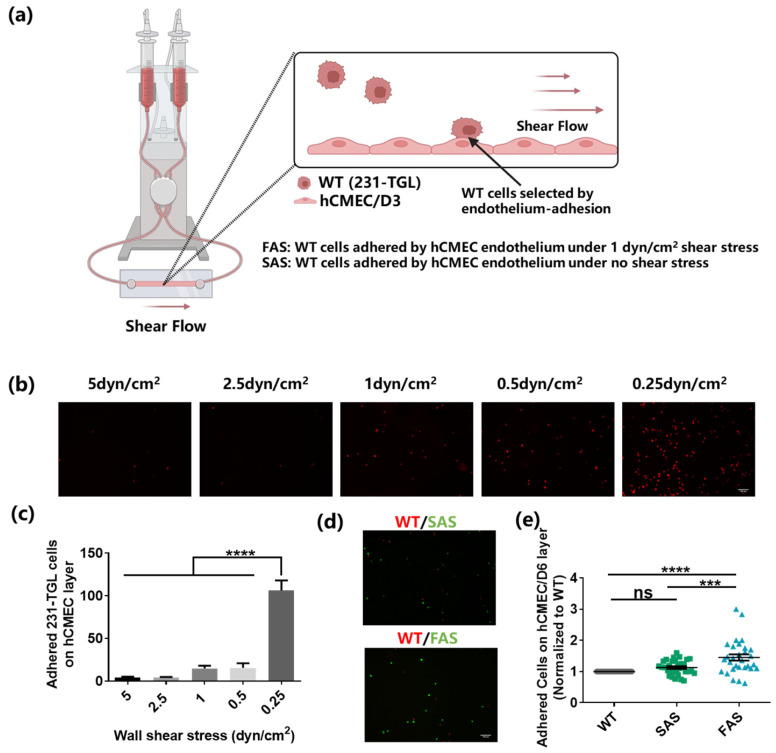
Brain-endothelium-adhesion-based selection of breast cancer cells. (**a**) The illustration of in vitro system for brain-endothelium-adhesion-based selection of breast cancer cells (created with biorender.com). (**b**,**c**) The adhesion of tumor cells to the brain endothelium under various levels of fluid shear stress. WT cells were labeled with red cell tracker and perfused into the flow chamber slides to adhere to the brain endothelium under 0.25, 0.5, 1, 2.5, and 5 dyn/cm^2^ shear stress for 15 min. The number of adhered cells was counted under a fluorescence microscope. Scale bar = 100 μm. *n* = 3. (**d**,**e**) The FAS cells exhibited enhanced adhesion ability under shear stress. WT cells (labeled with red cell tracker) and FAS cells or SAS cells (labeled with green cell tracker) were equally mixed and perfused into the flow chamber at the rate of 1 dyn/cm^2^ to adhere to the brain endothelium for 15 min. The number of adhered cells was counted under a microscope and normalized to the WT group. Scale bar = 100 μm. *n* = 3. All data are represented by mean ± SEM. The statistics among groups were analyzed using one-way ANOVA with the post hoc Bonferroni test. (ns: no significance, *** *p* < 0.001, **** *p* < 0.0001).

**Figure 2 ijms-24-07087-f002:**
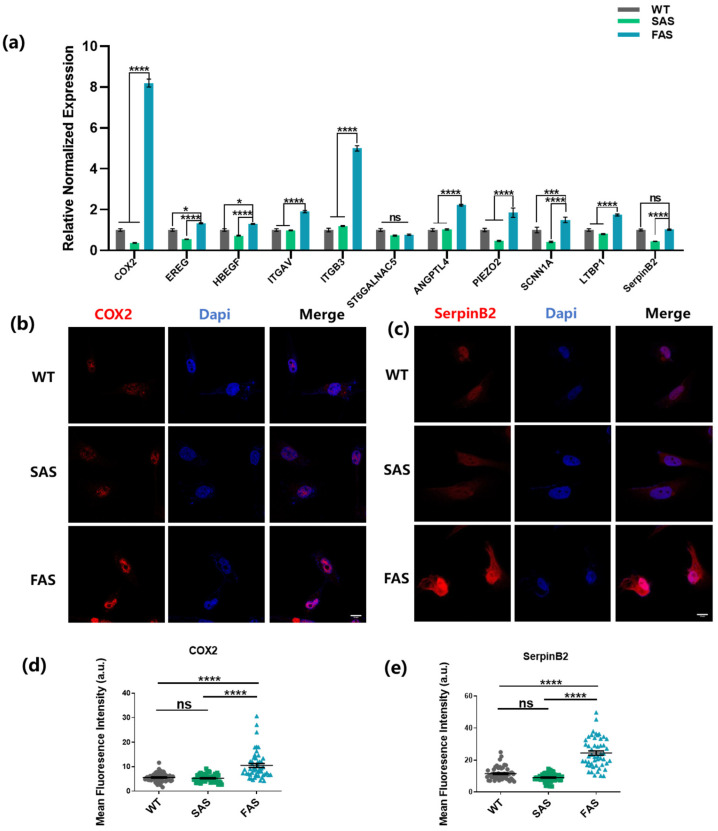
FAS cells up-regulate brain metastasis genes. (**a**) FAS cells showed enhanced expressions of brain metastasis-related genes. The gene expression was evaluated using qPCR. The statistics were calculated using two-way ANOVA with the post hoc Tukey test. *n* = 3. (**b**,**c**) The expressions of COX2 and SerpinB2 in all groups were tested using immunofluorescence staining. Scale bar = 10 μm. (**d**,**e**) Quantification of the fluorescence intensity of COX2 and SerpinB2 in (**b**,**c**). The statistics among three groups were calculated based on one-way ANOVA with the post hoc Bonferroni test. *n* = 50. (ns: no significance, * *p* < 0.05, *** *p* < 0.001, **** *p* < 0.0001).

**Figure 3 ijms-24-07087-f003:**
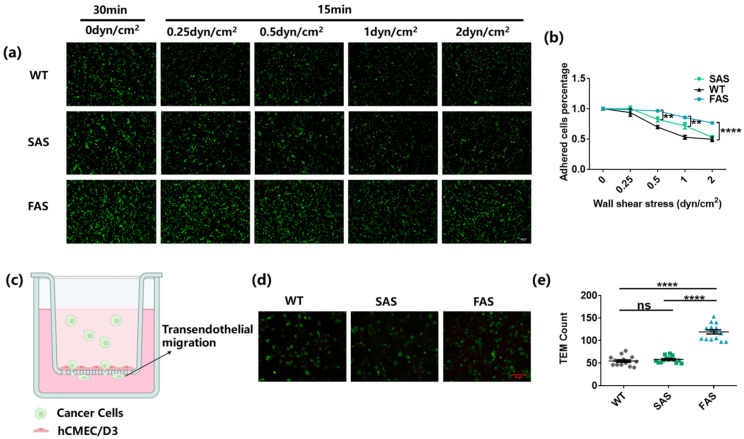
FAS cells exhibit enhanced adhesion to the brain endothelium and BBB transmigration ability. (**a**,**b**) FAS cells exhibited enhanced adhesion strength on the endothelium. All groups were labeled with green cell tracker and added into the flow chamber slides to adhere to the brain endothelium for 30 min. Then, 0.25, 0.5, 1, and 2 dyn/cm^2^ wall shear stress were applied to each slide for 15 min, respectively. The cancer cells remaining on the brain endothelium after each treatment of fluid shear stress were counted using the fluorescence microscope. Two-way ANOVA along with post hoc Tukey test were used to calculate the statistics. Scale bar = 100 μm. *n* = 3. (**c**) The illustration of trans-endothelial migration assay (created by biorender.com). (**d**,**e**) FAS cells exhibited enhanced BBB transmigration ability. An hCMEC/D6 monolayer was cultured on the top of the insert membrane and cancer cells were added to transmigrate through the monolayer to the lower chamber. The transmigrated cancer cells (marked by the green cell tracker) were imaged and counted using fluorescence microscopy. One-way ANOVA along with post hoc Bonferroni test was performed to analyze the statistics. Scale bar = 50 μm. *n* = 3. All data are represented by mean ± SEM. (ns: no significance, ** *p* < 0.01, **** *p* < 0.0001).

**Figure 4 ijms-24-07087-f004:**
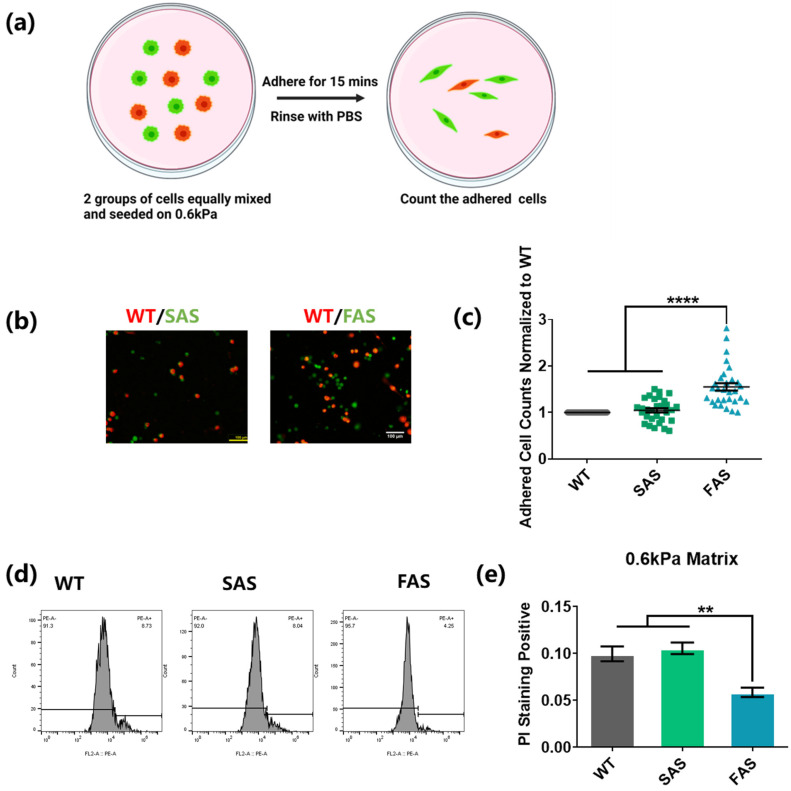
FAS cells exhibit advantages in cell adhesion and survival within the soft brain environment. (**a**–**c**) FAS cells had an enhanced adhesion ability on 0.6 kPa soft matrices. The same number of cancer cells from the WT group (labeled with red cell tracker) was mixed with cancer cells from the FAS group or the SAS group (labeled with green cell tracker) and seeded on the same 0.6 kPa polyacrylamide gels coated with collagen I. After 15 min, these cells were gently washed with PBS. This illustration was created using Biorender.com (**a**). The remaining cells were imaged and counted under fluorescence microscope. Scale bar = 100 μm. *n* = 3. (**d**,**e**) FAS cells exhibited lower cell apoptosis on soft matrices. All groups were seeded on 0.6 kPa polyacrylamide gels coated with collagen I in low-FBS medium overnight. The dead cells were then marked with PI and tested through flow cytometry. *n* = 3. All data are represented by mean ± SEM. The statistics among three groups were calculated based on one-way ANOVA with the post hoc Bonferroni test. (** *p* < 0.01, **** *p* < 0.0001).

**Figure 5 ijms-24-07087-f005:**
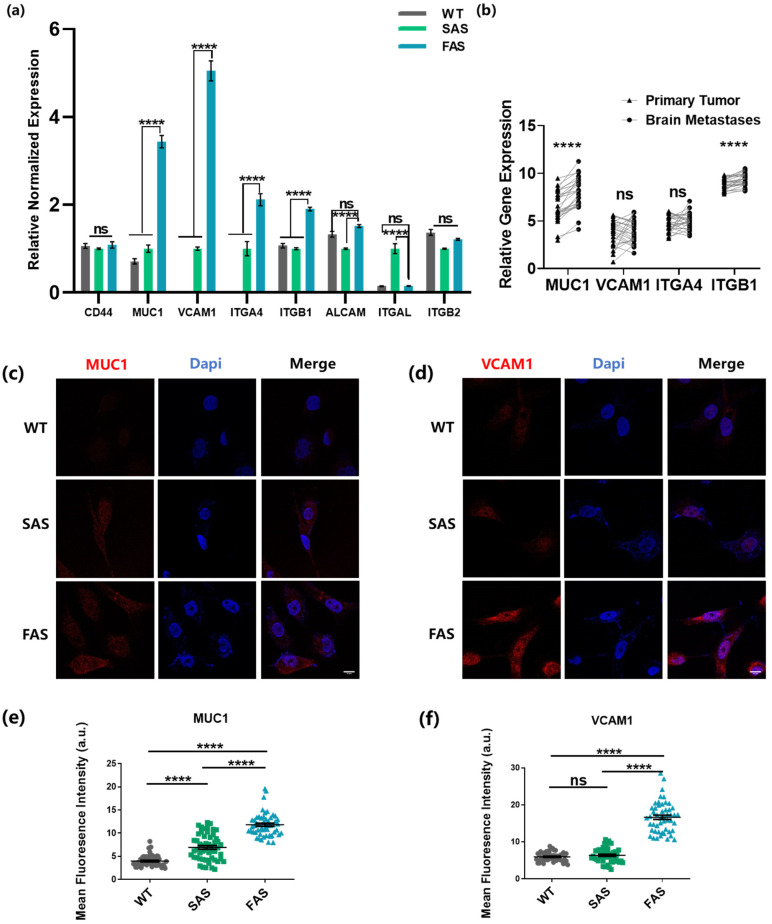
The FAS cells up-regulate multiple adhesion molecules that are clinically relevant to breast cancer brain metastasis. (**a**) FAS cells had enhanced expressions of brain metastasis-related adhesion genes. qPCR was conducted to test the gene expression. The statistics were analyzed based on two-way ANOVA along with the post hoc Tukey test. *n* = 3. (**b**) The highly expressed adhesion molecules in the FAS cells were clinically related to brain metastasis. The gene expression data were collected from the GEO public database (GSE173661, *n* = 25 samples). The paired comparison was analyzed using Graphpad. For each gene, a paired Student’s *t*-test was used to analyze the difference between the statistics in two groups. (**c**,**d**) The expressions of MUC1 and VCAM1 were tested through immunofluorescence staining. Scale bar = 10 μm. (**e**,**f**) Quantification of the expressions of MUC1 and VCAM1 in (**c**,**d**). The statistics among three groups were calculated based on one-way ANOVA with the post hoc Bonferroni test. *n* = 50. All data are represented by mean ± SEM. (ns: no significance, **** *p* < 0.0001).

## Data Availability

The dataset (GSE173661) analyzed in our study was collected from the public GEO database. https://www.ncbi.nlm.nih.gov/geo/query/acc.cgi?acc=GSE173661.

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
