# Peer review of "Adhesion to the Brain Endothelium Selects Breast Cancer Cells with Brain Metastasis Potential"

_ijms, 2023, doi:10.3390/ijms24087087_

Round 1

Reviewer 1 Report (Previous Reviewer 3)

I appreciate the authors for revising the manuscript according to my questions.

I think the manuscript is now acceptable.

Reviewer 2 Report (Previous Reviewer 1)

All prior comments have been addressed.

This manuscript is a resubmission of an earlier submission. The following is a list of the peer review reports and author responses from that submission.

Round 1

Reviewer 1 Report

This manuscript describes the establishment of a microfluidic system for the selection of shear-resistant breast cancer cells. The authors find that tumour cells selected as hyper-adherent under elevated shear stress demonstrate up-regulation of several adhesion markers previously shown to be associated with breast cancer metastasis in the literature, as well as increased proliferation, migration and survival.

The research provides an incremental contribution to the field that is worthy of publication. The study lacks some depth with respect to the discussion but also to substantiation of several claims which would be improved with further experiments. In particular, the authors state the desire to investigate changes specific to brain metastasis but this is insufficiently discussed and validated. While the scope in this area is limited, the authors have established flow parameters for future studies and provides a solid early dataset. Future work would achieve higher impact if microarray or proteomic studies are performed to assess novel expression changes.

Before acceptance for publication, the following should be addressed:

General:

1.       Spelling: Bonferroni not Bonforroni for statistical analysis; Tukey not Turkey for statistical analysis

2.       Magnification: Ensure to include magnification from both eyepiece lens (usually 10x) and objective lens when calculating magnification. The final magnification levels should be at least 10x what the authors state currently.

3.       Shear flow: Remove use of the word shear flow and ensure correct terminology is used e.g. “shear stress”.

Methodology

The authors must provide greater methodological detail before publication. The following should specifically be addressed:

1.       Please clarify how tumour cells are labelled with GFP or whether a specific cell source is used.

2.       Greater detail must be provided regarding the selection of shear stress resistant cells. For example: How are endothelial and tumour cells separated (that is, what markers used for separation) and how are the separated tumour cells assessed for purity (that is, lack of contamination with endothelial cells). Evidence can be provided in the supplementary section.

3.       Further detail must be provided to describe how adherent tumour cells are outgrown after selection? How are they trypsinised, cultured, passaged, how many passages before use, etc. It is not enough to reference publications 20,21, these are not similar papers by the same authors, so please elaborate.

4.       Please describe what “TGL” is? (section 4.2)

5.       Image J analysis – state which specific plugins or analysis used for each method.

6.       Provide detail on cell tracker addition or state whether the method was performed as described by the manufacturer.

7.       State which specific channels were used from Ibidi (catalogue number).

Results/Discussion

1.       The authors should discuss the first set of metastasis markers mentioned in section 2.2 and the significance of their expression or lack of expression (transferase and SerpinB2). In particular, for the latter genes, the transferase has greater association with the brain yet it is not up-regulated here. The authors should discuss if there are differences between studies that might account for this lack of induction of so-called brain markers, or hypothesise why this might be different in your study and what are the ramifications of this finding?

2.       Soft matrix made with polyacrylamide, it should be made clear in general text and in Figure 3 that this is collagen I binding dependent. This should be discussed in either the results or the discussion relevant to collagen attachment (alpha1 or alpha2 with beta 1) survival signals and how this concordant with literature and integrins identified as up-regulated in shear resistant clones.

3.       As noted for the methodology, it is unclear how long your endothelial cells are cultured under flow prior to tumour cell addition. Please make this very clear if they are pre-conditioned under flow and for how long. If they are not pre-conditioned, the authors must discuss how this might affect surface expression of endothelial ligands which they have stated are typically identified on brain EC (eg MUC, cadherin, VLA-4 and VCAM1). These and other adhesins/selectins are shear stress regulated and it is important to understand in the specific system and conditions used in this study whether or not these are actually expressed under the conditions generated. The authors must show with immunofluorescent staining whether these are expressed differentially in these endothelial cells at each shear stress level. Acute shear stress may induce release of Weibel Palade bodies but not longer term expression of other adhesins. The authors should also use these images to demonstrate cell morphology and whether cells remain confluent at each shear stress level.

4.       Further to above, not essential for this study, but use of blocking antibodies for each protein could be used to demonstrate the role of each ligand/receptor set in adhesion/migration.

5.       Earlier studies quoted in the reference section by the authors have carried out tumour cell selection and binding for 48h rather than for minutes.  Please discuss how selection without use of pre-conditioned endothelial cells may effect gene expression changes and selection and how this may lead to differences from other studies?

6.       VE-cadherin or N-cadherin are not investigated which are commonly up-regulated in metastatic cells as they allow direct connection with these more typical endothelial cadherins. It would be useful, although not essential to examine expression changes for these markers rather than E-cadherin in the tumour cells. At a minimum these markers should be discussed.

7.       Line 141-143 – “strength of stable adhesion between tumour cells between 2-4 dynes” – Unsure what the authors mean here. If you haven’t induced surface expression by pre-conditioning cells to a determined shear stress, this statement is potentially meaningless. Are these values and ranges consistent with earlier studies of shear stress and tumour cell adhesion to substantiate your statement? Discuss.

8.       Organotropism is frequently mentioned and the desire to understand or identify specific markers that allow brain uptake as opposed to lung, bone uptake etc. Does your shear stress rate reflect that appropriate to brain capillaries which harbor the BBB? Are our markers brain specific? Discuss.

9.       The authors discuss their future plans on other EC cells from other organs – the authors must expand their studies to ‘omics based study eg microarray, proteomics or surface proteomics. This would provide higher impact studies with novel expression profiles.

Reviewer 2 Report

Zhang et al. explored the interaction between breast cancer cells and brain endothelium cells for metastasis using ibidi flow chamber system in this study. They concluded that the “flow adhesion selected” (FAS) tumor cells highly expressed MUC1, VCAM1, VLA-4 and E-cadherin that could be associated with an increase of brain metastasis potential. However, several critical issues need to be clarified, particularly the characterization of FAS, before leading to a convincing conclusion.

1. The authors seem to use FAS to mimic the circulating tumor cells (CTCs) in this study. However, lack of characterization to support the features of FAS are similar to CTCs. Do both cells exhibit the same well-known biomarkers or similar functional properties?

2. Although the mRNA expression could be induced within several minutes (Fig. 2A), the alteration of protein expression (Fig. 1D) usually takes few hours after stimulation. The authors described that FAS was generated by 30 min of fluid shear stress and FAS showed a superior ability to adhere hCMEC/D3 in comparison with SAS or WT, implicating the change of adhesion molecules in the surface of FAS. Therefore, except mRNA, the authors should demonstrate that the critical adhesion molecules are indeed up-expressed in FAS. In addition, the effects of fluid shear stress on protein stability of the critical molecules should be clarified.

3. Since FAS is selected by 1 dyn/cm2 fluid shear stress, is it reasonable to compare cell adhesion ability between FAS and SAS under 1 dyn/cm2 condition (Fig. 1D)? The difference between both cells is unconvincing (Fig. 1E).

4. The authors validated many potential candidates responsible to brain metastasis based on previous studies. Which one is the most important for breast cancer brain metastasis? Or all of these molecules are indispensable? The causal regulations between fluid shear stress and the critical genes in tumor cell adhesion or other metastatic processes were not demonstrated. Therefore, knockout or knockdown of the target genes should be pursued to verify whether it could rescue fluid shear stress-induced metastasis.

Reviewer 3 Report

This manuscript describes the mechanism of adhsion between breast cancer cells and brain endothelium in brain metastasis. In particular, discovery fo gene expression associated with adhesion of breast cancer cells to the brain endothelium by rolling is of great interest. However, although the authors discovered gene expression related to brain metastasis, they did not provide sufficient experimental evidence for their gene's association with metastasis. Thus, it is not suitable for publication in IJMS journals.

Therefore, the authors should add some experiments.

1. The authors identified genes expressed only at the mRNA level during adhesion of breast cancer cells to the brain endothelium. In order to derive accurate results, validation at the protein level is required.

2. The authors compared genes highly expressed in FAS cells with clinical data, but no experimental validation was done for these genes. Therefore, the authors should confirm the association with the brain endothelium through a knock-out or knock-down model(cell line) of adhesion-related genes.

3. Since MDA-MB-231 cells are used in a brain metastasis mouse model, the authors can confirm the association between genes expressed from breast cancer cells and brain metastasis in a mouse model through a gene knock out cell model.

Round 2

Reviewer 2 Report

After revision, it is suitable for publication in IJMS journal.